# Local Non-Coding Regulatory Elements in Muscular Dystrophies

**DOI:** 10.3390/ijms26199690

**Published:** 2025-10-04

**Authors:** Harry Wilton-Clark, Sebastian Hernandez Rodriguez, Toshifumi Yokota

**Affiliations:** Department of Medical Genetics, University of Alberta, Edmonton, AB T6G 2R3, Canada; hwiltonc@ualberta.ca (H.W.-C.);

**Keywords:** ncRNA, lncRNA, miRNA, epigenetics, DMD, FSHD, LGMD, DM, CMD, OPMD

## Abstract

Muscular dystrophies are a class of diseases characterized by muscular weakness, breakdown, and heavily impaired function and quality of life. Numerous types of muscular dystrophies have been identified, with different causative genes and dystrophic mechanisms. While the majority of studies emphasize the protein product encoded by each gene, a growing body of research has identified non-coding elements as key regulators of muscular dystrophy. In this review, we summarize the common noncoding mechanisms known to regulate multiple forms of muscular dystrophies. We also highlight individual studies exploring local, disease-specific noncoding elements to each disease. Together, this provides a comprehensive overview of the major role of non-coding regulation in muscular dystrophies.

## 1. Background

Muscular dystrophies are a phenotypically linked group of genetic disorders, characterized by progressive muscular weakness and degeneration and typically distinguished from one another by their genetic etiology [1,2,3,4]. There are upwards of 30 known types of muscular dystrophies, and the number continues to expand as researchers identify an increasing number of causative mutations and pathological genes [1]. These varying types of diseases are generally subclassified into 9 general types: Duchenne muscular dystrophy (DMD), Becker muscular dystrophy (BMD), Myotonic dystrophy types I and II (DM1 and DM2), Facioscapulohumeral dystrophy (FSHD), Limb-Girdle muscular dystrophy (LGMD), Oculopharyngeal muscular dystrophy (OPMD), Emery-Dreifuss muscular Dystrophy (EDMD), and the Congenital muscular dystrophies (CMD) [2]. Although linked by a shared symptomology, these diseases arise from mutations in various unique genes and display a high degree of variability in disease progression.

Each muscular dystrophy primarily stems from impaired or defective protein expression from the causative gene for each disease. However, a growing body of research has also identified non-coding elements, including both non-coding RNAs (ncRNAs) and epigenetic regulation, as major contributors to muscular disease pathology [3,4,5]. ncRNAs are a class of RNA sequences that are transcribed from the DNA template but not translated into protein like messenger RNA would be [6,7]. Instead, ncRNA molecules play diverse roles in ribosomal function and protein translation, splicing, and gene expression regulation [7]. The ncRNAs subtypes typically associated with gene regulation and disease pathology are microRNAs (miRNAs), long non-coding RNAs (lncRNAs), short interfering RNAs (siRNAs), and PIWI-interacting RNAs (piRNAs) [8,9].

Both miRNAs and siRNAs regulate gene expression through a pathway known as RNA interference (RNAi). MiRNAs are small, single stranded RNA molecules with an average length of 22bp that generally act to suppress gene expression [10]. miRNAs regulate expression by associating with the RNA-induced silencing complex (RISC) and argonaute (AGO) proteins to form miRISC. Upon binding to a complementary messenger RNA (mRNA) sequence, miRISC induces either direct cleavage via AGO or inhibition of translation and subsequent degradation, with the specific mechanism dependent upon the overall target site complementarity [10,11,12]. In either case, expression of the target gene is downregulated. The target sites for miRNA are typically found in the 3′-UTR of mRNAs, and given their high tolerance for some mismatches, a single miRNA molecule can target hundreds of different mRNAs, acting as a master regulator for multiple genes and pathways [11,13]. siRNAs operate in a similar fashion to miRNAs, albeit with a few key differences. SiRNAs are double-stranded RNA molecules ranging from about 19–25bp, and also complex with RISC to mediate gene suppression [14,15]. In contrast with miRNAs however, siRNA target binding is extremely sequence-specific, and siRNAs typically only suppress a single mRNA transcript [14].

While both miRNAs and siRNAs are found body-wide, piRNAs play a more niche regulatory role and are only found in germline cells of mammals [16,17]. These single-stranded molecules are slightly longer than their RNAi counterparts, with lengths typically ranging from 26 to 31bp [18,19,20]. PiRNAs form a unique complex with specialized AGO proteins within the piwi subfamily, and are primarily implicated in recruiting RISC to suppress transposons within the germline [17,19]. Interestingly, rather than inducing cleavage or degradation, the piRNA-RISC complex has been shown to induce epigenetic silencing of target transposons by mediating H3K9 methylation, which forms heterochromatin [21,22].

The final class of regulatory ncRNAs are the lncRNAs, defined as being greater than 200bp in length with minimal to no coding activity [6,23]. These molecules display a wide variety of RNA-RNA, RNA-DNA, and RNA-protein interactions and act as master regulators, with a high level of both cell-specific and stage-specific expression [6,24]. In general, lncRNAs act as scaffolds or “hubs”, allowing for the binding of a variety of molecules to the same lncRNA. This can either facilitate the improved complexing of associated molecules to improve their function, or it can sequester molecules and prevent them from performing their normal function [23,25,26]. LncRNAs have even been shown to intercalate directly to double-stranded DNA inside the major groove, forming a RNA-DNA-DNA triple helix [27,28].

As our understanding of ncRNAs continues to deepen, so too does our understanding of their contributions to disease and disease progression. Aberrant ncRNA expression has been extensively explored in connection with muscular dystrophies, both as causative agents and as diagnostic and prognostic biomarkers [3,29,30,31]. In this review article, we aim to summarize the current state of knowledge regarding non-coding regulation of the muscular dystrophies.

## 2. Shared Non-Coding Regulation in the Muscular Dystrophies

Studies in both healthy and diseased muscle have identified a group of miRNAs known as the myogenic miRNAs (myomiRNAs) that have critical roles in muscle development, health, and function [32]. The most commonly recognized myomiRNAs in skeletal muscle disease are miR-1, miR-133a, miR-206, and less commonly miR-208, miR-486, and miR-499 [32,33,34].

MiR-1 targets both PAX7, a transcription factor that is crucial for growth and differentiation of the myogenic precursor satellite cells, and HDAC4, an epigenetic controller of numerous skeletal muscle genes [35,36,37,38]. By modulating these pathways, miR-1 is essential for skeletal muscle growth and function, independently of specific disease processes. Further, miR-1 levels have been found to be controlled by other muscle factors like MyoD, suggesting a delicate bi-directional regulation of this miRNA in muscle physiology [35,36,39]. Accordingly, miR-1 levels have been found in independent studies to be dysregulated compared to healthy tissues in nearly all muscular dystrophies except OPMD and CMD [5,30,40,41,42,43,44,45,46]. Interestingly, levels of miR-1 appear to be decreased in muscles but elevated in serum, suggesting possible utility as both a therapeutic target and as a serum biomarker [5]. MiR-133a has myriad known target mRNAs, but the most relevant target in muscle is serum response factor (SRF) [30,39,40,41,43,44,45,46,47,48,49]. SRF is a master transcriptional factor that regulates muscle growth, contraction, and sarcomere organization [50,51]. Similar to miR-1, miR-133a has been reported to be dysregulated in all dystrophies except CMD [30,40,41,42,43,44,45,46,49]. Lastly, miR-206 shares high sequence similarity and functional roles with miR-1, with an identical seed region that likely permits targeting of the same mRNAs [52,53]. Both molecules contribute to the regulation of myogenic differentiation by targeting PAX7 [52]. In addition, miR-206 notably regulates myogenic differentiation and myotube formation by targeting G6PD [52,54,55]. Consistent with the other major myomiRNAs, miR-206 is found to be dysregulated in all muscular dystrophies except OPMD and CMD [30,44,45,46,56].

In addition to the core myomiRNAs, several other miRNAs have been implicated in multiple muscular dystrophies. MiR-486-3p and miR-499 are both myogenically enriched and contribute to muscle proliferation and replication [57,58]. MiR-486-3p has been shown to be downregulated in DMD, FSHD, and LGMD, while miR-499 is upregulated in serum in DMD, FSHD, and CMD [43,59,60,61,62]. MiR-31, miR-195, and miR-199-3p are primarily known as tumor suppressors, yet all three have been independently found to be upregulated in muscle samples in FSHD and LGMD. Additionally, miR-31 and miR-199-3p were upregulated in muscles in DMD, but miR-31 was unchanged [46,63,64,65,66,67,68,69,70,71,72]. Finally, miR-233-3p, a molecule associated with inflammation and immune cell proliferation, has been shown to be elevated in muscles and muscle cell cultures in FSHD, LGMD, and DM type 1 [41,49,69,73,74]. Figure 1 depicts a summary of the commonly dysregulated miRNAs in muscular dystrophy, and whether they have been explored in relation to a given form of dystrophy (Figure 1).

The multi-disease dysregulation of these miRNAs suggests a generalized contribution to muscular dystrophy and muscle impairment, consistent with the patterns and mechanisms seen in the dysregulated myomiRNAs. However, many of these miRNAs have not been explored in relation to all types of muscular dystrophy, and there is high variability between diseases in the tissues tested for each miRNA (Figure 1). Future studies exploring the expression of each miRNA in each disease in both serum and muscle samples would be valuable, providing clarity towards the role of these miRNAs in disease. Furthermore, the specific mechanisms by which each miRNA contributes to disease remain unclear. Thorough characterization of the expression profile and mechanistic contributions of each miRNA to each dystrophy may yield novel insights into generalizable therapeutic mechanisms or biomarkers for disease progression. For example, transgenic overexpression of miR-486 has been shown to restore muscle health and strength in both a DMD mouse model and a muscle-impaired cancer mouse model [75,76]. However, despite the known dysregulation of miR-486 in FSHD and LGMD, no studies to date have explored the possible therapeutic benefit of miR-486 overexpression in these diseases. Given that miR-486 can reduce inflammation by targeting inflammatory cytokines such as IL-6, and can promote myogenesis by modulating the PTEN/AKT pathway, it would seem reasonable that miR-486 overexpression may have generalized benefit in many muscular dystrophies [57,75]. Continued exploration of both the expression levels and therapeutic potential of miR-486 across other muscular dystrophies could yield insight into miR-486 overexpression as a valuable therapy that works independently of specific disease mechanisms.

The expression pattern of these miRNAs also raises questions about distinguishing features between each muscular dystrophy. The expression of miRNA-195 was elevated in FSHD and LGMD muscles, but unexpectedly unchanged in DMD and DM1 muscles, and decreased in EDMD muscle culture. As miR-195 acts as a general inhibitor of muscle proliferation by targeting *Igfr1* and *Insr* and decreasing *MyoG* expression, it might be expected that similar direction of dysregulation would be observed in all muscular dystrophies, but this was not the case [77,78,79]. Continued studies into why the expression pattern of miR-195 and the other miRNAs differs between dystrophies could provide important insight into disease-specific pathways and features.

Finally, further studies could also explore the shared contributions of other non-coding regulatory elements beyond miRNA. For example, the lncRNA H19 is primarily known for its implications in cancer and tumorigenesis [80,81]. However, despite not being widely associated with muscular diseases, separate studies have found that H19 overexpression can alleviate both DMD and FSHD in mice, although neither study directly explored whether this affects tumorigenesis [82,83]. Further work assessing H19 and other shared lncRNA and epigenetic mechanisms in the muscular dystrophies has the potential to yield novel insights into disease progression and possible therapeutic approaches.

## 3. Non-Coding Regulatory Elements Local to Muscular Dystrophy Loci

In addition to the shared factors mentioned above, current research continues to identify disease-specific non-coding elements that are encoded directly within each causative gene. These elements include both ncRNAs as well as epigenetic modifications.

Epigenetic regulation includes any modification that alters chromatin state and conformation, modulating gene expression without changing the underlying DNA or producing any RNA [84]. These modifications include the post-translational modification of histones such as acetylation, methylation phosphorylation, or ubiquitination, as well as the direct methylation of CpG islands near the promoters of genes [84,85,86]. The effect on gene expression varies based on the type and location of the modification. Common epigenetic modifications known to increase gene expression via euchromatin formation include acetylation of the third histone at lysine residues 4 and 9 (H3K4 and H3K9), while H3K9 methylation is a common mark for suppression [84,87]. Here, we provide a summary of both the ncRNA and epigenetic regulatory elements found specifically within each muscular dystrophy.

### 3.1. Duchenne/Becker Muscular Dystrophy

Duchenne muscular dystrophy (DMD) is the most common form of muscular dystrophy, affecting 1/5000 male births and about 1/50,000,000 female births worldwide, and it is caused by mutations in the DMD gene encoding for the structural protein dystrophin [88,89,90]. DMD typically begins in the lower limbs of young children and progresses system-wide as patients age, proving fatal for most patients in their mid-late 20s as the disease progresses to the heart and diaphragm [91,92]. In some cases, a *DMD* mutation proves to have a substantially milder phenotype known as Becker muscular dystrophy (BMD) that does not manifest until much later in a patient’s life. While exceptions to this rule exist, in general it is thought that *DMD* mutations that disrupt the open reading frame (ORF) of the gene cause DMD, while mutations that preserve the ORF lead to BMD [93,94].

With over 2 million base pairs, *DMD* is the largest gene in the human genome [95]. Interestingly, less than 1% of the DMD sequence is part of the coding sequence (CDS) for dystrophin, with extremely large introns that could be heavily implicated in non-coding activity [96]. Bovolenta et al. explored this concept, characterizing the lncRNA expression profile of dystrophin from human brain, heart, and skeletal muscle tissues [97]. They found 14 novel lncRNAs encoded within *DMD*, 13 of which were located within introns and the final lncRNA found in the 3′ UTR [97]. The expression profile of these ncRNAs correlated with the expression of full-length dystrophin, suggesting that the coding and non-coding RNA products of *DMD* are similarly regulated. Furthermore, overexpression of several of the identified ncRNAs led to a reduction in full-length dystrophin, suggesting an internal regulation for dystrophin mediated by its own lncRNAs [97].

Epigenetic regulation has also been explored for DMD. Gherardi et al. found that two regions in *DMD*, located in intron 34 and exon 45, were heavily enriched in epigenetic marks favoring gene expression including demethylation of H3K4 and acetylation of H3K27 [98]. Further analysis revealed that the marked intron 34 region physically interacts with the dystrophin promoter to enhance gene expression, and that deletion of intron 34 leads to a reduction of overall dystrophin expression in BMD patients [98]. A separate study found that dystrophic mice show increased levels of *DMD* H3K9 trimethylation, a suppressive mark [99]. Treatment with a histone deacetylase inhibitor was able to reverse this effect, leading to a significant increase in the expression of dystrophin transcripts, but no assessment of functional benefit was performed. These findings clearly demonstrate diverse routes of noncoding regulation for DMD, shedding light on the wide phenotypic variability seen with different DMD-causing mutations.

### 3.2. Myotonic Muscular Dystrophy

Myotonic dystrophy type 1 (DM1) and type 2 (DM2) are autosomal dominant genetic diseases caused by nucleotide repeat expansions with a prevalence of 9.27 in 100,000 and 2.29 in 100,000, respectively [100]. Both myotonic dystrophies are multisystemic disorders affecting several parts of the body, including the eyes, skeletal muscle, and nervous system, where myotonia and muscle weakness are key features of the disease [101,102]. Although both diseases share similarities, they differ in muscle involvement, as DM2 primarily affects proximal muscles, whereas DM1 predominantly affects distal muscles [103]. Myotonic Dystrophies are characterized by the formation of RNA foci, referring to the accumulation of RNA complexes resulting from the transcription of expanded repeats [104]. RNA foci are particularly harmful because they can cause the sequestration of RNA-binding proteins, such as muscleblind-like (MBNL) proteins and other splicing factors, resulting in the dysregulation of various pathways and the development of spliceopathy [104]. DM1 arises from a trinucleotide (CTG)n expansion in the 3′ non-coding region of the *DMPK* gene, while DM2 is caused by uninterrupted tetranucleotide (CCTG)n expansion in intron 1 of the *CNBP* gene [101,102]. Moreover, unlike DM1, DM2 shows no correlation between the CCTG-repeat length and the severity of disease symptoms, and the disease does not worsen across generations [105].

DNA methylation and other epigenetic marks, including histone tail modifications such as acetylation and methylation, play a critical role in gene regulation and disease progression in DM [106,107]. In DM1, the methylation profile is linked with the four disease subtypes: congenital, childhood-, classic-, and late-onset [108]. In the congenital form of DM1 (CDM1), hyper-methylation was first reported in a segment upstream of the (CTG)n repeats between exons 11–15, preventing protein-DNA interactions, whereas this region remained un-methylated in less severe subtypes [109]. Subsequent studies also demonstrated that CTG repeats are flanked by two CTCF-binding sites hypothesized to function as insulator units, where CTCF binding mediates promoter–enhancer interactions and de-fines chromatin domains enriched with specific histone modifications [110,111]. However, in CDM1, repeat expansion results in CTCF-binding sites becoming hypermethylated, inhibiting CTCF binding and disrupting insulator function [110]. Moreover, such hypermethylation leads to the spread of heterochromatic epigenetic marks, such as H3K9me2 and affects the expression of the neighboring *SIX5* gene, a homeodomain transcription factor whose insufficiency contributes to the cataract phenotype [112,113,114]. These changes at the *DMPK* locus result in increased DMPK expression and accumulation of CUG-containing transcripts, potentially explaining the more severe phenotype and earlier onset observed in CDM1 [110]. However, this hypothesis remains contested as multiple lines of evidence challenge this model. For example, experiments conducted in DM1 transgenic mice have shown that CTCF binding is not affected by CTG-repeat length, despite the enrichment of heterochromatin epigenetic marks around the expanded CTG repeats, possibly due to the action of PCNA (proliferating cell nuclear antigen) [115]. In addition, another study showed that chromatin interactions (promoter-enhancer interactions) remain stable in different repeat lengths through 4C sequencing, suggesting it is unlikely that changes in chromatin interactions drive gene expression changes in DM1 [116].

Interestingly, changes in the methylation profile of *DMPK* are not exclusive to CDM1. Additional studies have determined that methylation abnormalities are largely cell- and tissue-specific, as well as age-specific, and do not necessarily correlate with the CTG repeat length of DM1 patients [117]. The hypermethylation downstream of the CTG repeats (CTCF binding site-2) also remains uncertain due to variable results and studies suggesting that the CpG-free expanded CTG tract may act as a barrier, preventing CpG methylation from spreading to downstream regions [117,118]. Other reports have shown that the methylation profiles of DMPK and neighbouring genes are cell-specific in WT samples [119]. For example, in myogenic cells specific hypermethylation occurs at CpG sites located between DMPK intron 2 and exon 4, as well as downstream of the DMPK promoter. Hypermethylation in the latter is associated with reduced CTCF binding at the DMPK promoter while enhancing binding at the hypomethylated 3′ end of the *DMWD* gene (located adjacent to the 3′ end of the DMPK gene) [119]. Thus, it has been proposed that this region could act as a methylation-sensitive and tissue-specific enhancer that drives DMPK expression in muscle cells [119]. Within this framework, another study using DM1 hESCs analyzed the methylation status of 23 up-stream and 11 downstream CpG sites of the CTG repeats to investigate methylation changes at the three stages of differentiation: hESCs, myoblasts, and myotubes [120]. No methylation was observed downstream of the CpG sites in all hESCs lines, whereas only cell lines carrying large repeats exhibited upstream methylation [120]. This supports previous observations that patients with larger expanded alleles are more likely to exhibit abnormal methylation, and patients with milder subtypes of DM1 rarely show high levels of methylation downstream the expand-ed repeat [113,118]. Moreover, the authors showed that the methylation levels upstream of the CTG repeat significantly increased during myogenic differentiation, but only in the lines with pre-existing methylation at their undifferentiated state [120]. This same effect was later observed in immortalized DM1 muscle lines derived from patients with different DM1 subtypes [121]. However, the authors did not observe changes in the differentiation capacity or expression of DMPK, SIX5, and DMWD genes between methylated and non-methylated cell lines, suggesting that hypermethylation of this region does not regulate the expression of DMPK and neighbouring genes during myogenesis [120].

Since methylation status and differentiation stage are linked in myogenic cells and methylation abnormalities are established in early stages, a recent study determined that epi-genetic abnormalities could be rescued after an early-stage complete excision of the CTG expansion cell [122]. The authors observed a switch from heterochromatin to euchromatin configuration in undifferentiated DM1 hESCs, but no change in the epigenetic configuration was observed in DM1-affected myoblasts and teratoma-derived fibroblasts after excision. Nevertheless, by generating induced pluripotent stem cells (iPSCs) from the edited CTG-deficient affected myoblasts, rescue was partially achieved, suggesting that myogenic differentiation plays a critical role in the epigenetic plasticity of cells [122]. Based on their findings the authors proposed a model in which methylation of CpG islands near the 3′-end of DMPK is driven by CTG repeat expansion and the recruitment of the de novo DNA methyltransferases (DNMT3a/b) to this region. Given the conflicting findings in the studies in this field, continued work is required to fully elucidate the relationship between CTG repeat expansions, epigenetic regulation, and DM1.

In addition to the epigenetic contributions, ncRNA-mediated local regulation of the *DMPK* has also been shown to play important roles in DM1. Various miRNAs have been identified as regulators, exhibiting both individual and cooperative effects [123]. Koscianska et al. found that miR-206/148a and miR-15b/16 could effectively repress DMPK transcripts, cooperatively regulating DMPK expression, as the simultaneous overexpression of both miRNAs pairs resulted in a significant decrease in DMPK mRNA [123]. In addition, the authors proposed that in DM1, the expanded DMPK transcripts may function as a “magnet” for miRNAs with CAG repeats (miR-16) in their seed sequences, preventing miRNA function. This proposed mechanism aligns with another study that found miR-107 function is impaired after DMPK sequestration, contributing to dysfunctional autophagy [124]. Lastly, while the specific contributions to DM1 remain unclear, the locally encoded lncRNA DM1-AS has been proposed to have a role in DM1 pathogenesis through the formation of toxic nuclear RNP aggregates and production of toxic repeat-associated non-ATG translation products [125]. A summary of non-coding elements regulating DM1 is presented in Figure 2.

Compared to DM1, the epigenetic contributions to DM2 are relatively uncharacterized. One study analyzed the methylation profile of two CpG islands between the 5′ promoter region and the 3′ end of the CCTG repeats of the *CNBP* gene [107,126]. The authors found that while the CpG islands at the 5′ end showed hypomethylation, and the CpG island at the 3′ end was hypermethylated, no changes existed between DM2 patients and healthy controls, suggesting that repeat expansion does not influence the methylation status [126]. In addition, unlike DM1, the *CNBP* methylation profile was not tissue-specific, as no difference was observed between blood and muscle samples. Overall, these results suggest that the local *CNBP* methylation profile does not play a primary role in the pathogenesis of DM2. Lastly, Malatesta et al. investigated cell senescence indices in satellite cell-derived myoblasts from DM2 patients [4,127]. The authors reported increased heterochromatin in DM2 myoblasts, consistent with senescent cells, with heterochromatin accumulation occurring earlier in DM2 satellite cell-derived myoblasts than in healthy controls. However, the specific regions and the molecular mechanism behind this observation remain unclear, indicating further research is needed to understand the influence of expanded (CCTG)n repeats on the chromatin status in DM2.

### 3.3. Facioscapulohumeral Dystrophy

Facioscapulohumeral dystrophy (FSHD) is an autosomal dominant disorder arising from inappropriate expression of the *DUX4* gene found within the D4Z4 genetic locus [128,129]. Though the disease mechanism is not fully understood, it is attributed to the pathogenic expression of the transcription factor DUX4, which is primarily expressed during embryonic development and not in the muscles of healthy tissue [130]. Clinically, FSHD is most often an adult-onset disease characterized by dystrophy of the muscles in the face, scapula, and humerus [128]. Although it is rarely fatal, FHSD can impose significant disability and deterioration in quality-of-life for affected patients [130].

Numerous ncRNAs have been found to be encoded within the D4Z4 locus that are thought to directly contribute to aberrant *DUX4* expression. Cabianca et al. identified a novel lncRNA called DBE-T that was encoded in the D4Z4 locus and selectively expressed in FSHD patients [131]. DBE-T was demonstrated to lead to the de-repression of the D4Z4 region containing *DUX4* by recruiting Ash1L, a histone methyltransferase that demethylates the silencing H3K36 mark that is present in healthy cells. Not only was DBE-T only overexpressed in FSHD cells, but it also only functioned when expressed in *cis* to the D4Z4 region, suggesting a precise noncoding mechanism regulating FSHD [131]. A separate study identified several miRNA-sized fragments produced in the D4Z4 locus, however their contribution to FSHD pathology was unknown [132]. Analysis of an independent dataset revealed expression of miRNAs with matching sequences in cancer and stem cells, but no further functional validation was performed [132]. In a later study, the same authors found that treatment with exogenous siRNAs targeting the same region as the identified miRNAs epigenetically suppressed *DUX4* expression [133]. Specifically, siRNA treatment led to an increase in D4Z4 H3K9 dimethylation, suggesting a critical role played by endogenous miRNAs in normal *DUX4* silencing. Finally, Block et al. identified substantial levels of antisense transcription of D4Z4 that does not correlate with a protein product, and the authors theorized that this may lead to the production of currently uncharacterized ncRNAs [134].

### 3.4. Limb-Girdle Muscular Dystrophy

The limb-girdle muscular dystrophies (LGMDs) are a group of more than 30 genetically heterogeneous disorders characterized by progressive proximal or distal muscle weak-ness and atrophy, primarily affecting the pelvic and shoulder girdle muscles [135,136]. LGMDs are autosomal inherited; however, this large compendium of disorders is classified into two types: LGMD-Ds (previously LGMD type 1), which are dominantly inherited, and LGMD-Rs (previously LGMD type 2), which are recessively inherited [135,137]. Nevertheless, within the classification, there is still heterogeneity as each of the subtypes impacts different genes. LGMD-Ds affect a variety of genes going from ribonucleoproteins (hnRN-PDL), nuclear transporters (transportin 3), ECM proteins (collagen 6), and others [135,137]. On the other hand, LGMD-Rs mainly affects sarcoglycans (α-Δ sarcoglycan), enzymes involved in the glycosylation of dystroglycans (fukutin), dysferlin (involved in vesicle trafficking regulation), and other structural proteins of the sarcolemma. Consequently, although some subtypes share similar pathology, there is a broad spectrum of clinical presentations among patients, making it challenging to diagnose and predict their clinical outcomes without genetic testing [138].

Epigenetic modifications have been explored at *DYSF*, the LGMD R2 dysferlin-related causal locus. A study exploring the *DYSF* methylation profile of three CpG islands (CpGi33, CpGi64, CpGi93), 4 CpG island shores, and CpGs located upstream from the regulatory region of the DYSF gene revealed no changes between the methylation profiles of patients, carriers, or healthy volunteers [139]. In all groups, CpGi64, CpGi93 and R2 regulatory regions were completely unmethylated, whereas CpGi33 and R1 regulatory regions were hypermethylated [139]. This data suggests that DNA methylation status does not play a critical role in *DYSF* expression, and that different mechanisms likely drive *DYSF* downregulation.

Contrasting with this result, another study that investigated the methylation status of *DYSF* promoter using MDRE (Methylation-Sensitive Restriction Enzymes)-qPCR in atherosclerotic cardiovascular disease patients found that hypermethylation of the *DYSF* promoter was positively related with *DYSF* expression [140]. Additionally, the knockdown and overexpression of DNMT1 in THP-1 cells demonstrated that the *DYSF* promoter was hypomethylated in the knock-down cells and hypermethylated in the overexpressing cells [140]. At the same time, knockdown cells showed reduced *DYSF* transcripts while overexpressing cells showed increased DYSF mRNA levels. As a result, the authors proposed that hypermethylation was actually an activating epigenetic mark in the *DYSF* promoter, possibly by blocking the binding of repressive transcription factors or inducing the recognition by methylation-dependent transcription factors [140]. Discrepancies between these studies could be a result of different assays used to study the methylation profile of *DYSF*. However, because other genes, such as DMPK, exhibit tissue-specific epigenetic marks, further studies are needed in skeletal muscle samples from LGMD R2 dysferlin-related patients to characterize whether changes in methylation and chromatin status occur during the disease and their implications in pathogenesis.

### 3.5. Emery-Dreifuss Muscular Dystrophy

Emery-Dreifuss Muscular Dystrophy (EDMD) is an extremely rare muscular dystrophy, affecting between 1/100,000–1/250,000 people [141,142]. Mutations in multiple genes can cause EDMD, including X-linked *EMD* or *FHL1* mutations and autosomal dominant *LMNA*, *SYNE1*, or *SYNE2* mutations. The classic clinical presentation of EDMD includes a triad of contractures, cardiac abnormalities, and muscular weakness, typically beginning in the upper limbs [143,144].

Likely due to its low prevalence, very few studies have explored noncoding elements in the genes affiliated with EDMD when compared with the more common dystrophies such as DMD or DM. One study found that a lncRNA encoded on the antisense strand of *SYNE1*, SYNE1-AS1, contributes to cardiac hypertrophy by sequestering miR-525-5p [145]. Interestingly, knockdown of SYNE1-AS1 led to a reduction in hypertrophic markers in cell culture, suggesting both a mechanism and therapeutic target for the cardiac dysfunction seen in *SYNE1*-related EDMD.

Interestingly, although the studies were not directly related to EDMD, mutations in the *LMNA* gene are associated with epigenetic dysregulation due to associations with lamina-associated domains (LADs). LADs are heterochromatic (H3K9me2 and H3K9me3-rich) regions at the periphery of the nucleus that associate with lamin and play a role in gene repression [146]. However, as lamin production is affected in EDMD, this leads to abnormal gene expression and chromatin organization [147]. In line with these mechanisms, a study involving human induced pluripotent stem cells (hiPSCs) with mutations from a patient diagnosed with dilated cardiomyopathy found that *LMNA* mutations lead to abnormal gene expression [148]. This occurs because the mutations disrupt interactions between the nuclear lamina and chromatin regions, which in turn causes an abnormal nuclear structure [148]. Building on the understanding that *LMNA* disruption affects local epigenetics, a recent study identified 17 differentially methylated CpG sites within the *LMNA* gene in patients with fetal congenital heart disease (CHD) compared to healthy controls [149]. From the CpGs identified, five were located on the 3′ UTR and between the LMNA promoter and the first exon, while the remainder were located across the gene body. Therefore, the authors suggested that, as LMNA transcripts are alternatively spliced to produce lamin A or lamin C, methylation could be a regulatory mechanism to balance the expression of these isoforms [149]. Further studies must be conducted to fully elucidate whether methylation is relevant for lamin alternative splicing and EDMD pathogenesis, as specific mutations have been reported to perturb mRNA splicing, leading to the accumulation of a mutant prelamin A in cells [150]. In this case, the hypermethylation of CpG located on the LMNA promoter could explain the prior reports of low LMNA levels in CHD patients by repressing transcription [149]. However, because LMNA mutations can result in different disease phenotypes, further studies are needed in diagnosed EDMD patients to understand the importance of epigenetic changes in the *LMNA* gene that affect the disease phenotype.

To the best of the authors knowledge, no studies to date have explored local noncoding elements in *EMD*, *FHL1*, or *SYNE2* and their relation to EDMD.

### 3.6. Oculopharyngeal Muscular Dystrophy

Oculopharyngeal Muscular Dystrophy (OPMD) is a dystrophy causing adult-onset weakness in the muscles of the eyelids and around the throat, leading to drooping eyelids and difficulty swallowing [151,152]. While the general prevalence is relatively low at about 1/100,000, two key groups demonstrate vastly elevated rates of the disease due to a founder effect. French-Canadians in Quebec show an OPMD prevalence of about 1/1000, and Bukhara Jews living in Israel have an even higher prevalence of about 1/600 [151,153]. OPMD is caused by mutations in the *PABPN1* gene, which is important for the regulation of PolyA tail length during mRNA processing [151].

Working in HeLa cells, Adbelmohsen et al. identified a circular ncRNA encoded within the *PABPN1* locus called circPABPN1 [154]. CircPABPN1 was found to associate with and sequester the RNA-binding protein HuR, preventing HuR from binding to PABPN1 mRNA and suppressing its translation [154]. While not directly linked to OPMD, this study suggests a possible ncRNA-mediated regulatory mechanism contributing to the disease.

More recently, Shademan et al. found that a non-coding isoform of PABPN1 was overexpressed following an alternative polyadenylation event commonly seen in cases of OPMD [155]. Although the function of this ncRNA is unknown, the authors theorize that it may serve as a regulatory molecule for the expression of other RNAs.

### 3.7. Congenital Muscular Dystrophies

Lastly, Congenital Muscular Dystrophy (CMD) is a continuously growing class of autosomal recessive dystrophies characterized by their early onset muscle weakness detectable at birth or early infancy [156,157]. Upwards of 30 genes are implicated in CMD, and the clinical phenotype and severity vary greatly based on the causative gene [158]. The combined prevalence of all CMDs is less than 1/100,000, and given the low prevalence and high number of disease-causing genes, very few studies have been performed assessing noncoding elements in the context of CMD [158].

An unrelated breast cancer study found that *LAMA2*, a gene causing a form of CMD known as Ullrich congenital muscular dystrophy (UCMD), can be suppressed through lncRNA-mediated hypermethylation of the *LAMA2* promoter [159]. While it is unknown if the same lncRNA is implicated in UCMD, it may represent a shared mechanism and therapeutic target for aberrant *LAMA2* expression in both diseases.

To the best of the authors knowledge, no studies to date have explored local noncoding elements in any other gene associated with CMD.

## 4. Conclusions

Non-coding elements, both ncRNAs and epigenetic modifications, play a critical role in the regulation of muscular dystrophies. However, these non-coding contributions are often understudied compared to their coding counterparts. Here, we provide an overview of both shared and unique non-coding mechanisms regulating the different muscular dystrophies. We summarize commonly dysregulated miRNAs across all dystrophies, highlighting their critical role for muscle health and function, and identify gaps in our current knowledge. We also review the current state of knowledge regarding the local non-coding elements at each genetic locus causing muscular dystrophy, highlighting the unique dysregulation that can lead to each form of the disease. This study acts as a consolidated foundation from which further studies can be based, paving the way for future studies and therapies based around non-coding elements in muscular dystrophies.

## Figures and Tables

**Figure 1 ijms-26-09690-f001:**
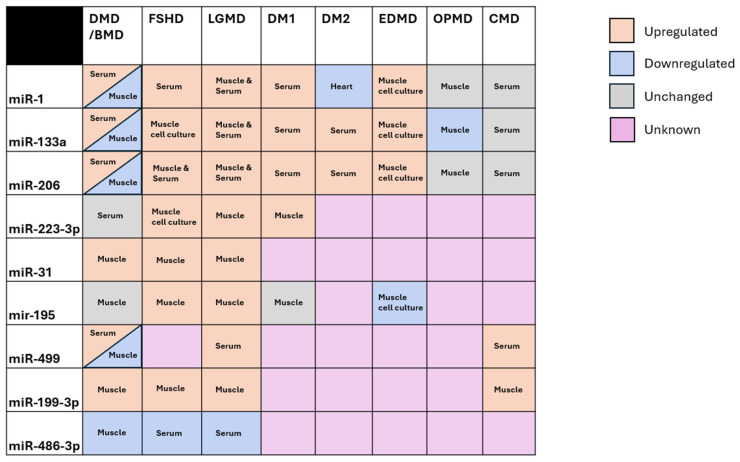
A depiction of whether commonly dysregulated miRNAs are dysregulated in each form of muscular dystrophy, and the tissue used for assessment. Orange boxes represent upregulation in that disease, blue boxes represent downregulation, and grey boxes represent normal expression. Pink boxes represent that to the best of the authors knowledge, no studies have explored this miRNA-disease combination. All subtypes of LGMDs were included in the LGMD column, as most pertinent studies did not separate by subtype.

**Figure 2 ijms-26-09690-f002:**
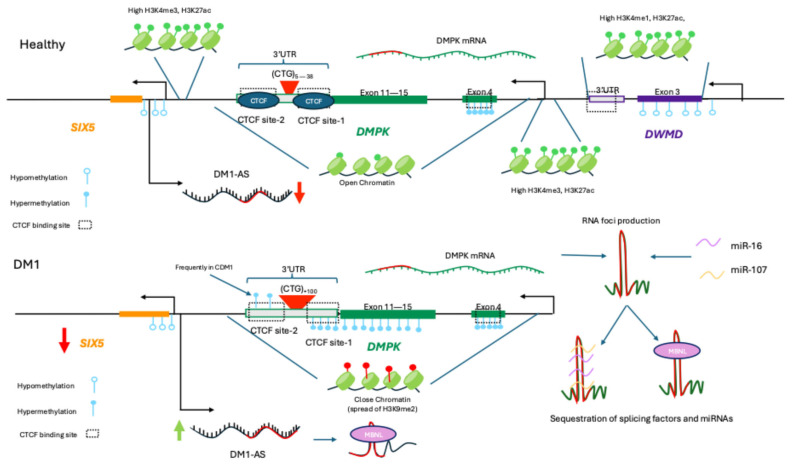
Schematic representation of chromatin and methylation profile in healthy and DM1 affected muscle cells. Black arrows indicate transcription start site of each gene or antisense transcript. Red arrows indicate decrease in gene expression and green arrows increase in gene expression. In healthy condition DMPK gene remains unmethylated with the exception exon 4 and surrounding introns. In addition, unmethylated CTCF binding sites upstream (CTCF1) and downstream (CTCF2) the CTG repeats allows CTCF binding. SIX5 and DWMD promoter regions are characterized by open chromatin status. DWMD 3′ end as enhancer of DMPK driving increase expression in muscle cells. DM1 antisense transcripts (DM1-AS) is in low levels in WT cells. However, in DM1, hypermethylation of CTCF sites and upstream regions prevents CTCF to bind, driving downstream effects that are still debated. In addition, hypermethylation results in a reduction in expression of SIX5 (homeodomain transcription factor) contributing to the pathogenesis of DM1. Moreover, histone tail modifications changes to inactive marks that close chromatin. DM1 antisense transcripts (DM1-AS) carrying expanded CTG repeats increase in DM1 hypothesized to be involved in the pathogenesis of DM1. Finally, DMPK transcripts function as a “magnet” for splicing factors (MBNL) and miRNAs with CAG repeats (miR-16 and miR-107) in their seed sequences, preventing miRNA’s normal function.

## Data Availability

No new data was generated as part of this review article. All referenced data is available from the original publications, as listed in the references section.

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
