# Peer review of "Local Non-Coding Regulatory Elements in Muscular Dystrophies"

_ijms, 2025, doi:10.3390/ijms26199690_

Round 1

Reviewer 1 Report

Comments and Suggestions for Authors

The manuscript "Local non-coding regulatory elements in Muscular Dystrpophies" represents an important contribution to the field.  A review of this topic is very timely and needed to help researchers asses the state of the field.  The authors are complimented on their organization and high quality writing.  Section 3.2 was a little difficult to wade through but overall very well written manuscript on a cutting edge topic.

I have a few concerns before publication:

In section 2 when discussing miRNA's on line 121, that paragraph uses the term dysregulation 6 times and it is used elsewhere.  But that offers very little insight to the biology of the disease.  It would be easy to say that 1000's of genes are dysregulated in a severe disease like muscular dystrophy but that doesn't actually tell us anything about what it means or what could researchers look at next or what could be a therapy.  The closest it comes is lines 152 - 156 when discussing possible therapies as "massive utility" and "therapeutic benefit".  But how?  By what mechanisms would this work?  More detail is needed here, maybe a paragraph to attempt to work out some of the molecular details.  Give specifics.

Line 214 it mentions "increase in DMD transcript expression".  Expression of what?  Transcripts in general?  Dystrophin?  Something else?  Did the mice actually improve their phenotype?

Figure 2 is very detailed, which is good, but a little confusing.  Can the names of the genes like Six5 and DMPK be put in a different font so they stand out as obvious markers of gene names and put them in consistent locations in both images?  It made the figure more difficult to interpret.  Also the miR's, it's not entirely clear from the figure or legend what is going on with those, what are they doing?  Finally in this regard, the text would benefit from a little background on the function of Six5 and perhaps the family of genes that it comes from.  The section 3.2 itself is a bit cumbersome and I wonder if it could be shortened to be a little more straight forward.

Also, what about SNPs?  it seems like these could become very important in every single one of these non-coding elements!  Has anyone investigated this?

In general it could be mentioned that we need to know more about how all of these work at the mechanistic level.  So many of these are just "dysregulated" and we don't really know what they mean.

Author Response

Reviewer 1: (Comments underlined, responses in blue)

The manuscript "Local non-coding regulatory elements in Muscular Dystrpophies" represents an important contribution to the field.  A review of this topic is very timely and needed to help researchers asses the state of the field.  The authors are complimented on their organization and high quality writing.  Section 3.2 was a little difficult to wade through but overall very well written manuscript on a cutting edge topic.

Thank you for taking the time to review and provide feedback on our manuscript.

In section 2 when discussing miRNA's on line 121, that paragraph uses the term dysregulation 6 times and it is used elsewhere.  But that offers very little insight to the biology of the disease.  It would be easy to say that 1000's of genes are dysregulated in a severe disease like muscular dystrophy but that doesn't actually tell us anything about what it means or what could researchers look at next or what could be a therapy.  The closest it comes is lines 152 - 156 when discussing possible therapies as "massive utility" and "therapeutic benefit".  But how?  By what mechanisms would this work?  More detail is needed here, maybe a paragraph to attempt to work out some of the molecular details.  Give specifics.

Information has been added throughout this section to ensure a higher level of detail, and Figure 1 has also been updated to better reflect specific changes in miRNA expression. Both the paragraph and the figure now mention the direction of dysregulation, as well as the tissue that dysregulation was observed in.

Additionally, the paragraph on line 121 now provides a higher level of speculative detail regarding specific mechanisms, and better highlights the need for continued research on these miRNAs. It now reads:

” The multi-disease dysregulation of these miRNAs suggests a generalized contribution to muscular dystrophy and muscle impairment, consistent with the patterns and mechanisms seen in the dysregulated myomiRNAs. However, many of these miRNAs have not been explored in relation to all types of muscular dystrophy, and there is high variability between diseases in the tissues tested for each miRNA (Figure 1). Future studies exploring the expression of each miRNA in each disease in both serum and muscle samples would be valuable, providing clarity towards the role of these miRNAs in disease. Furthermore, the specific mechanisms by which each miRNA contributes to disease re-main unclear. Thorough characterization of the expression profile and mechanistic contributions of each miRNA to each dystrophy may yield novel insights into generalizable therapeutic mechanisms or biomarkers for disease progression. For example, transgenic overexpression of miR-486 has been shown to restore muscle health and strength in both a DMD mouse model and a muscle-impaired cancer mouse model [82,83]. However, de-spite the known dysregulation of miR-486 in FSHD and LGMD, no studies to date have explored the possible therapeutic benefit of miR-486 overexpression in these diseases. Given that miR-486 can reduce inflammation by targeting inflammatory cytokines such as IL-6, and can promote myogenesis by modulating the PTEN/AKT pathway, it would seem reasonable that miR-486 overexpression may have generalized benefit in many muscular dystrophies [64,82]. Continued exploration of both the expression levels and therapeutic potential of miR-486 across other muscular dystrophies could yield insight into miR-486 overexpression as a valuable therapy that works independently of specific disease mechanisms.

The expression pattern of these miRNAs also raises questions about distinguishing features between each muscular dystrophy. The expression of miRNA-195 was elevated in FSHD and LGMD muscles, but unexpectedly unchanged in DMD and DM1 muscles, and decreased in EDMD muscle culture. As miR-195 acts as a general inhibitor of muscle proliferation by targeting Igfr1 and Insr and decreasing MyoG expression, it might be expected that similar direction of dysregulation would be observed in all muscular dystrophies, but this was not the case [84–86]. Continued studies into why the expression pattern of miR-195 and the other miRNAs differs between dystrophies could provide important in-sight into disease-specific pathways and features.

Finally, further studies could also explore the shared contributions of other non-coding regulatory elements beyond miRNA. For example, the lncRNA H19 is primarily known for its implications in cancer and tumorigenesis [87,88]. However, despite not being widely associated with muscular diseases, separate studies have found that H19 overexpression can alleviate both DMD and FSHD in mice, although neither study directly explored whether this affects tumorigenesis [89,90]. Further work assessing H19 and other shared lncRNA and epigenetic mechanisms in the muscular dystrophies has the potential to yield novel insights into disease progression and possible therapeutic approaches.”

Line 214 it mentions "increase in DMD transcript expression".  Expression of what?  Transcripts in general?  Dystrophin?  Something else?  Did the mice actually improve their phenotype?

This section now reads:

” Treatment with a histone deacetylase inhibitor was able to reverse this effect, leading to a significant increase in the expression of dystrophin transcripts, but no assessment of functional benefit was performed.”

Figure 2 is very detailed, which is good, but a little confusing.  Can the names of the genes like Six5 and DMPK be put in a different font so they stand out as obvious markers of gene names and put them in consistent locations in both images?  It made the figure more difficult to interpret.  Also the miR's, it's not entirely clear from the figure or legend what is going on with those, what are they doing?  Finally in this regard, the text would benefit from a little background on the function of Six5 and perhaps the family of genes that it comes from.  The section 3.2 itself is a bit cumbersome and I wonder if it could be shortened to be a little more straight forward.

As per your insightful suggestions, the figure has been updated to improve clarity, and additional discussion has been placed in the text to better explain some elements of the figure. Furthermore, section 3.2 was simplified and summarized where possible to make it less cumbersome.

Also, what about SNPs?  it seems like these could become very important in every single one of these non-coding elements!  Has anyone investigated this?

This is a very insightful point, and SNPs in non-coding elements could certainly alter their expression and effectiveness. To the best of the authors knowledge, no studies have explored SNPs in these regions in the context of muscular dystrophies.

In general it could be mentioned that we need to know more about how all of these work at the mechanistic level.  So many of these are just "dysregulated" and we don't really know what they mean.

The lack of detailed information available has been further highlighted. The section now reads “However, many of these miRNAs have not been explored in relation to all types of muscular dystrophy, and there is high variability between diseases in the tissues tested for each miRNA (Figure 1). Furthermore, the specific mechanisms by which each miRNA contributes to disease remain unclear. Thorough characterization of the expression profile and mechanistic contributions of each miRNA to each dystrophy may yield novel insights into generalizable therapeutic mechanisms or biomarkers for disease progression.”

Reviewer 2 Report

Comments and Suggestions for Authors

Line 114: Please indicate what the functional similarities are that miR-206 has with miR-1.

Line 114: MiR-206 'M’ should not be capitalized.

Line 124, 127 and 129:  There are several types of LGMD, please indicate which subtypes are affected/included in the study. Well done for specifying DM type 1 in line 129.

Lines 130 and 132: Figure 2, should it be written Figure 1? Figure 1: instead of dysregulated, can you specify upregulated and downregulated?

Line 161: since H19 is related to tumorigenesis, does the article show if overexpressing in DMD and FSHD mice increases probability the have a cancer?

Line 184: replace ‘boys’ with ‘males’ to indicate biological sex, not gender. Also females can be affected, at a lower rate (1: 50 000 000) https://doi.org/10.1016/j.slsci.2016.07.004 Line 360 : summarize the paragraph in 2-3 lines.

The DM section is long and pertinent, but a summary would be useful for the readers.

Line 394: reference numbers do not match. [135] does not mention LGMDs. Verify when the unmatching starts and ends.

Line 396: use the recent nomenclature for LGMDs : LGMD1(letter) and 2(letter) were replaced with LGMDR(number) (recessive) and LGMDD(number) (dominant).  See articles: https://doi.org/10.1016/j.nmd.2018.05.007 and 10.3390/jcm12144769 You can keep the previous nomenclure in parenthesis.

Line 391 and + : Some protein names are capitalized and others are not.

Line 408 : dysflerlin should be written dysferlin

Author Response

Reviewer 2:

Line 114: Please indicate what the functional similarities are that miR-206 has with miR-1.

Thank you for taking the time to review and provide feedback on our manuscript. This section now reads “Lastly, miR-206 shares high sequence similarity and functional roles with miR-1, with an identical seed region that likely permits targeting of the same mRNAs [59,60]. Both molecules contribute to the regulation of myogenic differentiation by targeting PAX7 [59]. In addition, miR-206 notably regulates myogenic differentiation and myotube formation by targeting G6PD [59,61,62].”

Line 114: MiR-206 'M’ should not be capitalized.

The erroneous capitalization has been removed and now reads “Lastly, miR-206…”

Line 124, 127 and 129:  There are several types of LGMD, please indicate which subtypes are affected/included in the study. Well done for specifying DM type 1 in line 129.

The figure caption now reads “All subtypes of LGMDs were included in the LGMD column, as most pertinent studies did not separate by subtype.”

Lines 130 and 132: Figure 2, should it be written Figure 1? Figure 1: instead of dysregulated, can you specify upregulated and downregulated?

The incorrect figure attrition has been corrected, and the figure itself has been reworked to include both the direction of dysregulation, and the tissue in which it was measured. Please note that the MS word function “Show changes” may need to be temporarily disabled to properly display the new figure.

Line 161: since H19 is related to tumorigenesis, does the article show if overexpressing in DMD and FSHD mice increases probability the have a cancer?

Neither article directly explores tumorigenesis, which has been noted. This section now reads “However, despite not being widely associated with muscular diseases, separate studies have found that H19 overexpression can alleviate both DMD and FSHD in mice, although neither study directly explored whether this affects tumorigenesis”

Line 184: replace ‘boys’ with ‘males’ to indicate biological sex, not gender. Also females can be affected, at a lower rate (1: 50 000 000) https://doi.org/10.1016/j.slsci.2016.07.004

This section now reads “Duchenne muscular dystrophy (DMD) is the most common form of muscular dystrophy, affecting 1/5000 male births and about 1/50,000,000 female births worldwide, and caused by mutations in the DMD gene encoding for the structural protein dystrophin”, and the suggested reference has been incorporated.

Line 360 : summarize the paragraph in 2-3 lines.

Section 3.2 has been generally simplified and summarized throughout to make the section less cumbersome.

The DM section is long and pertinent, but a summary would be useful for the readers.

Section 3.2 has been generally simplified and summarized throughout to make the section less cumbersome.

Line 394: reference numbers do not match. [135] does not mention LGMDs. Verify when the unmatching starts and ends.

Thank you for catching this error. Reference unmatching was resolved and verified.

Line 396: use the recent nomenclature for LGMDs : LGMD1(letter) and 2(letter) were replaced with LGMDR(number) (recessive) and LGMDD(number) (dominant).  See articles: https://doi.org/10.1016/j.nmd.2018.05.007 and 10.3390/jcm12144769 You can keep the previous nomenclure in parenthesis.

Thank you for this clarification. The section now reads: “LGMDs are autosomal inherited; however, this large compendium of disorders is classified into two types: LGMD-Ds (previously LGMD type 1), which are dominantly inherited, and LGMD-Rs (previously LGMD type 2), which are recessively inherited”. The more recent nomenclature has been incorporated into the rest of the manuscript as well.

Line 391 and + : Some protein names are capitalized and others are not.

Protein names have been standardized for consistency.

Line 408 : dysflerlin should be written dysferlin

This typo has been fixed.

Correction References:

We confirm that all references included in the manuscript are relevant and important to the information included in the text.